# Exploring the High Variability of Vegetative Desiccation Tolerance in Pteridophytes

**DOI:** 10.3390/plants11091222

**Published:** 2022-04-30

**Authors:** Gerardo Alejo-Jacuinde, Luis Herrera-Estrella

**Affiliations:** 1Department of Plant and Soil Science, Institute of Genomics for Crop Abiotic Stress Tolerance (IGCAST), Texas Tech University, Lubbock, TX 79409, USA; galejoja@ttu.edu; 2National Laboratory of Genomics for Biodiversity (Langebio), Centro de Investigación y de Estudios Avazados del Instituto Politécnico Nacional, Irapuato 36824, Mexico

**Keywords:** pteridophytes, ferns, *Selaginella*, desiccation tolerance, protection mechanisms

## Abstract

In the context of plant evolution, pteridophytes, which is comprised of lycophytes and ferns, occupy an intermediate position between bryophytes and seed plants, sharing characteristics with both groups. Pteridophytes is a highly diverse group of plant species that occupy a wide range of habitats including ecosystems with extreme climatic conditions. There is a significant number of pteridophytes that can tolerate desiccation by temporarily arresting their metabolism in the dry state and reactivating it upon rehydration. Desiccation-tolerant pteridophytes exhibit a strategy that appears to be intermediate between the constitutive and inducible desiccation tolerance (DT) mechanisms observed in bryophytes and angiosperms, respectively. In this review, we first describe the incidence and anatomical diversity of desiccation-tolerant pteridophytes and discuss recent advances on the origin of DT in vascular plants. Then, we summarize the highly diverse adaptations and mechanisms exhibited by this group and describe how some of these plants could exhibit tolerance to multiple types of abiotic stress. Research on the evolution and regulation of DT in different lineages is crucial to understand how plants have adapted to extreme environments. Thus, in the current scenario of climate change, the knowledge of the whole landscape of DT strategies is of vital importance as a potential basis to improve plant abiotic stress tolerance.

## 1. Introduction

Drought stress is one of the major causes of crop productivity loss that will be aggravated by climate change. For decades, the understanding of the adaptive mechanisms of plants to tolerate drought, but also desiccation, has been of great interest because of its importance in plant evolution and potential for crop improvement. Plants resistant to desiccation are commonly known as “resurrection plants” (vascular and non-vascular plants) due to their remarkable ability to come back to life from an apparent lifeless condition. The attribute behind resurrection is denominated as desiccation tolerance (DT), which can be described as the ability to reverse arrest metabolism after cells have reached an air-dry state [1]. A leaf water potential of about −100 MPa (approximately equivalent to air of 50% relative humidity at 20 °C) is the accepted standard threshold that a plant must survive to be considered as a desiccation-tolerant species [2,3]. This water potential nearly corresponds to a water content of 0.1 g H_2_O/g dry weight, which is insufficient to maintain a monolayer of water around macromolecules and thus metabolism stops [4]. Desiccation kills most plants, but resurrection species limit cellular damage to a repairable level and can successfully recover biological activity upon rehydration [5]. The ability to survive desiccation is a complex trait that relies on an orchestrated regulation of physiological, biochemical, and molecular process.

During the early evolution of land plants, primitive plants evolved mechanisms of DT that allowed them to survive increasingly longer periods of dryness in terrestrial ecosystems. Therefore, the acquisition of DT was crucial for the colonization of land environments [6]. These primitive plants probably possessed constitutive protection mechanisms coupled with an active cellular repair process, similar to those observed in many desiccation-tolerant bryophytes [7]. A constitutive protection strategy is inferred from the fact that some bryophytes can survive rapid water loss, even when desiccation is reached in a few minutes [6,8]. This DT strategy involves the constitutive expression of genes involved in cellular protection, and the constant presence of proteins, enzymes, and metabolites is required to tolerate desiccation. Constitutive mechanisms for DT have an implicit fitness cost caused by the constant production of protective components and compounds required for DT. Although it is generally accepted that desiccation-tolerant bryophytes employ a constitutive strategy, there is evidence that some also have inducible protection mechanisms [9]. Most likely due to their high energetic and metabolic costs, constitutive DT in vegetative tissue was lost during the early evolution of tracheophytes and recruited into reproductive structures such as seeds and spores to secure dispersal and survival [3,10]. Since constitutive DT in vegetative tissues was lost early during the evolution of tracheophytes, the VDT observed in tracheophyte lineages of this monophyletic plant division originated from multiple independent events indicatives of convergent evolution [11].

In contrast to bryophytes, most desiccation-tolerant vascular plants typically dry out on a longer time scale (from several hours to days) [12], and cannot survive a rapid water loss that some bryophytes can tolerate. The strategy of plants that survive desiccation only if water loss is gradual is mainly based on inducible dehydration tolerance mechanisms [6]. However, there is evidence that the VDT strategy in early diverging tracheophytes (i.e., pteridophytes) is not restricted to inducible mechanisms, but a constitutive component has also been reported in some species. In agreement with their phylogenetic position, VDT in pteridophytes (lycophytes and ferns; Figure 1) has previously been proposed as an intermediate mechanism between the constitutive mechanisms exhibited by bryophytes and the inducible response of desiccation-tolerant angiosperms [6]. However, desiccation response in pteridophytes is highly variable, and knowledge on the responses to desiccation in this group is crucial to decipher the convergent evolutionary origin of this ability. Pteridophyte species reported as desiccation-tolerant plants include the lycophyte *Selaginella* and ferns of the genus *Anemia*, *Cheilanthes*, *Hymenophyllum*, *Pellaea*, and *Polypodium*, among others. In this review, we discuss the diversity of desiccation-tolerant pteridophytes, their high plasticity in the adaptations and protection mechanisms of VDT as well as some particularities in their desiccation response. Finally, we discuss the multiple abiotic stress tolerance that some desiccation-tolerant plants exhibit and its possible potential for improving crop abiotic stress tolerance.

## 2. Incidence, Habitat, and Anatomical Diversity of Desiccation-Tolerant Pteridophyte Species

Although VDT occurs in most major groups of plants, it is present in a small proportion of species compared to the total estimated flora. A recent review compiled a list of the land plants that exhibit VDT and reported that this ability is present in approximately 600 species so far, with representative members in all major lineages of land plants except for gymnosperms [11]. Species that exhibit VDT comprise around 1.14% of the estimated number of bryophytes, 0.91% of pteridophytes, and 0.08% of angiosperms [13,14]. Therefore, the incidence of VDT has been previously described as relatively common in bryophytes, infrequent in pteridophytes, and rare in angiosperm species [15]. However, VDT in a large proportion of bryophytes and pteridophytes has not yet been determined, thus the incidence of this trait in these groups could be significantly higher than the percentages reported here. A large number of pteridophytes including both lycophytes and ferns are adapted to xeric conditions and probably possess DT ability. For example, there are reports of field observations of xerophytic ferns that curl their fronds during drought periods and show vigorous leaves a short time after rainfall [16]. Estimations of the total number of ferns that could possess VDT ranges between 700 to 1000 species including a considerable proportion of filmy ferns of which most if not all are desiccation-tolerant [17]. Additionally, many *Selaginella* species belonging to the subgenus *Tetragonostachys* (estimated to contains at least 45 species) occupy deserts of southwestern North America, being clearly adapted to extreme environmental conditions [18]. However, evidence of VDT has only been reported for less than a quarter of the *Tetragonostachys* members [19]. A recent study which included some *Tetragonostachys* species showed that VDT is even present in a species adapted to very moist habitats, suggesting that likely all species within this subgenus are tolerant to desiccation [20].

The ability to survive desiccation is also present in *Isoetes*, commonly known as quillworts, but knowledge on their responses to desiccation and mechanisms of DT have been little studied. Although *Isoetes* species most often occupy aquatic or semiaquatic habitats, some experience occasional drought. In contrast to other resurrection pteridophytes, the VDT in *Isoetes* is apparently restricted to corms [21], underground stem structures that acts as a food-storage structure. For instance, the desiccated corms of *Isoetes taiwanensis* can remain viable for several months, and a few days after rehydration, they produce new leaves and roots [22]. Due to the seasonal availability of water in the habitats of some *Isoetes*, it is possible that several of these species could display VDT ability. Characterization of VDT in pteridophytes has been hampered by the lack of simple methods to determine whether this trait can be applied under field conditions and without the need to sacrifice a whole individual. Recently, simple and reproducible methods using leaf explants that can be easily used in the field to determine VDT have been reported [20,23]. Although angiosperms are undoubtedly the most successful plants, being the dominant flora on our planet, the potential to evolve VDT seems to be at least ten times higher in pteridophyte species. This is an important reason to encourage research in characterizing the molecular and biochemical processes involved in the DT of pteridophytes compared to other plant lineages.

Although desiccation-tolerant plants can occupy a wide range of environments, most inhabit inselbergs, described as monolithic rock outcrops poorly covered by soil [24]. Inselbergs are subjected to extreme environmental conditions including rapid fluctuations in water availability. Therefore, inselbergs have been proposed as centers of diversity for desiccation-tolerant vascular plants (Figure 2) [24]. Desiccation-tolerant organisms are frequently found growing together in inselbergs. Rapid fluctuation in water availability in these microhabitats make them inhospitable for most plants, and desiccation-tolerant species represent the dominant vegetation in these sites. However, as indicated above, desiccation-tolerant plants are not restricted to arid environments and can also be found in humid habitats (e.g., *Lindernia brevidens*, which is endemic to the montane rainforest [25]). In fact, the canopy of tropical and temperate forests displays a great diversity of desiccation-tolerant species with a substantial number of epiphytic ferns (mainly of the Hymenophyllaceae family) [17]. Water availability within the tree canopy is highly variable, thus represents suitable sites to be colonized by desiccation-tolerant organisms.

In contrast to bryophytes that can be rehydrated by dew or rain, desiccation-tolerant vascular plants seem to rehydrate only after receiving rain [15,26]. Some desiccation tolerant pteridophyte species can occupy desert regions with as little as 20 to 50 cm of total precipitation per year [27]. Therefore, to restore a metabolically active state when favorable conditions arise, these plants must be very efficient in water acquisition. Foliar water uptake represents a common feature shared between desiccation-tolerant pteridophytes and angiosperms. Leaf structures with a role in water uptake in ferns include trichomes, hairs, and scales [16], but their function is not limited to water uptake, but also to prevent rapid water influx that could produce cellular damage [26]. A study in the desiccation-tolerant fern *Pleopeltis polypodioides* determined that scales contribute to general water management during the desiccation process [28]. Besides improving water absorption during the rehydration stage, frond scales also have a role during dehydration by preventing rapid water loss. Furthermore, an in situ study showed that leaf irrigation is insufficient for the recovery of the entire plant. When only the leaves of the ferns *Pentagramma tirangularis* and *Pellaea andromedifoli* receive water, the stele and roots remain desiccated [29]. This study suggests that in natural conditions, recovery after desiccation involves not only foliar water uptake but also root turgor and capillary action.

Apparently, there are no mandatory anatomical characteristics for VDT in pteridophytes, and these species can show contrasting anatomical and morphological characteristics. Specifically, desiccation-tolerant ferns can be divided in two groups: (1) ferns that show a well-developed cuticle and includes several families, and (2) filmy ferns, mainly represented by Hymenophyllaceae species, with a very simple frond structure composed of one or a few cell layers with a rudimentary or absent cuticle [30,31]. Filmy ferns are generally associated with microenvironments subjected to recurrent desiccation and rewatering cycles. In contrast to ferns with well-developed cuticle, filmy ferns do not have an efficient mechanism to retain water and can lose water within minutes when exposed to dry air [16]. Most of the characteristics exhibited by filmy ferns suggest that the DT ability in this group of plants is mainly conferred by a constitutive strategy, rather than the widespread inducible response observed in tracheophytes. An additional similarity of filmy ferns to bryophytes is their rapid rehydration, as some of these species fully recovered the hydrated state after one hour of soaking in water [31]. Overall, the desiccation characteristics of filmy ferns appear to be more related to that of bryophytes than those of vascular plants. However, few studies have shown that inducible mechanisms can also occur in filmy ferns. For instance, *Crepidomanes inopinatum* increases the concentration of soluble sugars and activity of superoxide dismutase (SOD) in response to dehydration [30].

## 3. Overview of Recent Insights about the Origin of VDT in Vascular Plants

The ability to tolerate desiccation is a common and widespread feature of plant reproductive structures. Although the ability to tolerate extreme dehydration in vegetative tissues of angiosperms is rare, most seeds can survive desiccation (known as orthodox seeds). Even though some angiosperms produce desiccation-sensitive seeds (recalcitrant seeds), the pollen of such species display DT [32]. Spores are the reproductive structure of non-seed plants, and they also exhibit DT. Tolerance to extremely low water contents (equilibrium with ~1% RH) has been reported for some fern spores, but they can display a huge variation in their longevity in the desiccated state, ranging from a few days to several months [33]. Therefore, apparently all vascular plants possess the genetic potential for DT, but in most species, DT is restricted to reproductive structures.

In contrast to the constitutive or inducible protection strategies displayed in vegetative tissues, the DT of a reproductive structure such as orthodox seeds is developmentally regulated. During the last stage of seed development, denominated seed maturation, orthodox seeds acquire DT [34]. The acquisition of DT during seed development is associated with multiple cellular processes including the accumulation of carbohydrates and late embryogenesis abundant (LEA) [35]. The hormone abscisic acid (ABA), which mediates the response to several environmental stresses (e.g., drought, salinity, cold), also participates in the induction of dormancy and DT during seed maturation [36]. The embryos of orthodox seeds generally lose their DT during germination. Interestingly, there is a short time window when the germinated seeds of some plants can re-activate DT when treated with ABA, indicating a major role of this hormone in the regulation of DT [34].

The protection mechanisms exhibited in vegetative tissues of angiosperms resemble those observed in orthodox seeds. Due to such similarities, previous studies have suggested that VDT in angiosperms evolved from the re-activation of the developmental-seed program [6,37,38,39]. Analysis of the genome of the resurrection monocot *Xerophyta viscosa* supports the notion of the activation of seed pathways in VDT [40]. The genome of this species has a significant expansion of some LEA families, and specifically genes encoding the LEA 4 family are accumulated during drying and rehydration. Interestingly, the promoters of LEA 4 members in *X. viscosa* have a significant enrichment of the DNA binding motif for the transcription factor ABI5, an ABA-responsive transcription factor that participates in seed development. These results suggest that VDT in this resurrection angiosperm could have arisen from the reactivation of regulatory networks such as those activated during the maturation of orthodox seeds.

A rewiring event of pre-existing seed pathways has been proposed as the origin of VDT in angiosperms including the reactivation of seed development master regulators in vegetative tissues. Seed maturation is mainly controlled by a set of transcription factors named the LAFL network (LEC1, ABI3, FUS3, and LEC2) and the specific expression of some of these regulators in the reproductive structures of non-seed plants suggests a conserved role in spore development [41]. Although the function of these transcription factors has been primarily associated with reproductive structures, there is evidence that some of them also participate in VDT. The moss *Physcomitrella patens* can survive desiccation if incubated with ABA, but this desiccation-tolerant phenotype is dependent of the presence of ABI3 genes [42]. Interestingly, a co-expression analysis in *X. viscosa* determined a partial conservation of the *Arabidopsis* ABI3 regulon, indicating a shared regulatory network between seeds and VDT [40]. A further analysis determined that desiccation tolerant *Xerophyta* species have at least four paralogs of the ABI3 gene, but only one is induced during both seed maturation and VDT [43]. However, this ABI3 gene expressed in response to vegetative desiccation in *Xerophyta* lacks the B3 domain required for binding to its cognate binding site in the promoters of some of its target genes. Furthermore, most of the putative target genes of ABI3 in *Xerophyta viscosa* did not show the motif recognized by the B3 domain (i.e., the RY *cis-*acting element) [43]. Then, the role of this specific ABI3 paralogue in VDT could likely be via the interaction with other transcription factors.

Current knowledge in resurrection angiosperms strongly suggests that VDT evolved from the reactivation of protective mechanisms of orthodox seeds. Thus, this reactivation of seed pathways in vegetative tissues have been proposed by means of conserved seed regulators including LAFL transcription factors. Comparative analysis between seed developmental stages and vegetative tissue in response to desiccation in the same species could determine if this hypothesis is correct. The characterization of the gene expression changes during seed maturation in a desiccation-tolerant plant has only been performed in the monocot *Xerophyta humilis* [43]. Although this study also determined a significant overlap in the induced genes during seed development and during the dehydration of vegetative tissue, the canonical LAFL was only induced during seed maturation but not in vegetative tissues. Therefore, this observation indicates that the activation of DT in vegetative tissue is the result of the rewiring of the original regulatory network activating DT in *X. humilis* seeds.

A similar scenario regarding DT in reproductive structures (i.e., spores) and its reactivation in vegetative tissues of pteridophytes has also been proposed. Unfortunately, to our knowledge, there are no studies characterizing global gene expression programs during spore maturation in resurrection pteridophyte species. Since processes that take place during spore development are similar to those observed in seeds, it has been proposed that the maturation phase shares some similarities between both reproductive structures [44]. However, it is important to consider that some of the major events described as related to DT acquisition in seeds may not be present during spore development. For example, late maturation in seeds is associated with chlorophyll degradation because its retention could be detrimental to longevity in the dry state [45]. In contrast, several fern species produce chlorophyllous spores, which could be associated with a rapid recovery of photosynthesis at the gametophytic stage [46]. Further studies are required to determine the key protection mechanisms for DT in spores and the regulatory networks controlling such processes to gain insights into the origin of VDT in pteridophytes.

## 4. Intermediate DT Strategy in Pteridophytes: Constitutive and Inducible Responses

Plants can experience different types of cellular damage associated with mechanical, structural, or metabolic stresses during desiccation [5,37]. Desiccation-tolerant species possess several mechanisms to avoid or minimize the detrimental effects of desiccation. The most common and widespread VDT mechanisms described in vascular plants include increased antioxidant levels (enzymatic and non-enzymatic antioxidants), changes in lipid composition, adjustment of carbohydrate metabolism, modifications in cell wall properties, induction of early light-induced proteins (ELIPs), accumulation of members of the LEA, and small heat shock proteins (sHSPs) families, among others [47,48,49,50]. An extensive survey of the main findings of transcriptomic studies identified a core of desiccation responses apparently conserved among all green plants, suggesting that VDT was derived from the mechanisms found in ancestral land plants [51]. The transcriptional response displayed by resurrection plants showed a set of universally utilized strategies for VDT including the accumulation of LEA, ELIPs, sHSPs, compatible solutes, and antioxidants. However, the same authors pointed out that plants have additional species-specific mechanisms to acquire VDT.

The intermediate position of pteridophytes in the land plant phylogeny correlated with their VDT strategy (Figure 1), which shares some features with the dehydration-inducible protection response observed in angiosperms and others with the constitutive mechanisms present in bryophytes. As discussed previously, desiccation-tolerant pteridophytes display highly diverse and even contrasting morphological adaptations. The presence of leaf structures to diminish water loss during dehydration suggest a dehydration-induced VDT strategy. On the other hand, a relatively simple leaf structure with no adaptations to retain water indicates a major constitutive component for VDT. Here, we compiled increasing evidence that some pteridophyte species with a predominantly inducible VDT strategy could also show a constitutive component and vice versa.

The photosynthetic apparatus is very liable to injury during water stress; therefore, resurrection plants have several mechanisms to prevent irreversible damage [52]. Desiccation-tolerant plants can present a strategy to preserve the photosynthetic machinery at the desiccated state (homoiochlorophyllous), allowing for fast recovery during rehydration, or alternatively, a strategy to dismantle the photosynthetic apparatus during dehydration and resynthesize and reassemble it upon rehydration (poikilochlorophyllous) [53]. In poikilochlorophyllous, degradation of chlorophyll and dismantling of chloroplast significantly reduce ROS formation, but reconstruction of the photosynthetic apparatus upon rehydration results in longer recovery times compared to the homoiochlorophyllous species. Homoiochlorophyllous plants are associated with habitats with rapid alternations of wet and dry cycles, whereas poikilochlorophyllous evolved in habitats where plants remain in a desiccated state for 8–10 months [54]. Poikilochlorophylly is restricted to monocot species in which the recovery of full metabolic activity can take up to 72 h after rehydration [55].

Desiccation-tolerant pteridophytes are classified as homoiochlorophyllous species [11]. Therefore, their chloroplast ultrastructure (including stromal thylakoid, grana thylakoid and chloroplast membrane system) remains intact at the desiccated state [56], allowing for recovery of full photosynthetic activity a few hours after rehydration. Most homoiochlorophyllous species are subjected to a partial loss of chlorophyll during desiccation, but never exceeding critical levels. Nevertheless, the fern *Pleopeltis pleopeltifolia* loses more than two-thirds of its chlorophyll during desiccation and despite only recovering half of the initial chlorophyll content 24 h after rehydration, its photosynthetic activity has similar values to the fully hydrated conditions indicating rapid recovery [57]. Fast recovery after desiccation is only possible in homoichlorophyllous species, but the loss of most chlorophyll in this fern contrasts such a strategy. A similar pattern was also observed in the filmy ferns *Hymenoglossum cruentum* and *Hymenophyllum dentatum,* which experience a significant decrease in chlorophyll pigments during a dehydration–rehydration cycle, but not complete dismantling of the photosynthetic apparatus [31]. The authors of this last study proposed an intermediate strategy between homoiochlorophylly and poikilochlorophylly in these filmy ferns. Interestingly, desiccation-tolerant corms of *Isoetes* can neither be classified as homoiochlorophyllous nor poikilochlorophyllous because its leaves (and also roots) decay and lose function after desiccation, and their corms develop completely new photosynthetic tissue upon rehydration [22]. The way that these species have evolved to deal with desiccation increases the diversity of VDT strategies observed in pteridophytes. Furthermore, *I. taiwanensis* also exhibits crassulacean acid metabolism (CAM) [58], providing the opportunity to study the crosstalk between a strategy to improve water-use efficiency, and the ability to tolerate extreme dehydration.

Although a significant number of pteridophytes have been described as desiccation-tolerant species, few studies have been carried out to characterize their desiccation response. Among the desiccation-tolerant pteridophytes, the best characterized species is the lycophyte *Selaginella lepidophylla* [59]. Our knowledge on this species includes physiological studies to determine photosynthetic characteristics [60,61], ultrastructural changes at the desiccated state [62,63], early analysis of gene expression [64], metabolic profiling [65], and a sequenced genome [66]. However, desiccation-tolerant pteridophytes display diverse responses to desiccation and the knowledge of VDT in this group is still limited. Therefore, we present an overview of the VDT mechanisms in pteridophytes as determined by physiological and biochemical approaches (Table 1) as well as those identified by metabolic, proteomic, and transcriptomic analysis (Table 2).

As observed in Table 1, several studies in pteridophytes determined an increase in soluble sugars during desiccation. The accumulation of sugars has an essential role in VDT participating in the formation of intracellular glasses and replacing water molecules [86]. Nevertheless, the type and proportions of carbohydrates accumulated during hydrated conditions or in response to dehydration are highly variable among desiccation-tolerant plants [50] and even species-specific adaptations might exist. Accumulation of sucrose is a widespread response in plants that display VDT, and at least in angiosperms, represents the major carbohydrate in desiccated tissues [47]. Upon dehydration, only sucrose is accumulated in the desiccation-tolerant grass *Eragrostis nindensis* compared to some desiccation-sensitive relatives [39], indicating the importance of this sugar in VDT. Sucrose has also been reported in some desiccation-tolerant pteridophytes but with different accumulation patterns. While in most pteridophyte species the concentration of sucrose increases in response to desiccation, some studies have reported an opposite pattern. For example, the lycophyte *Selaginella bryopteris* exhibits a significant decrease in its sucrose content in desiccated and rehydrated states [75], however, a key role of sucrose in VDT cannot be discarded, as in this species, sucrose probably accumulates under normal conditions and is then converted into other protective sugars during dehydration. Furthermore, although *S. lepidophylla* exhibits several dehydration-induced responses (Table 1 and Table 2), this species also accumulated high levels of sugars including sucrose in a constitutive fashion [83]. Maintaining high sucrose levels regardless of hydration status is one of the VDT strategies observed in desiccation-tolerant mosses [87]. This is one example of an intermediate VDT strategy in a desiccation-tolerant pteridophyte that employs both dehydration-induced and constitutive mechanisms.

Mechanisms for ROS scavenging could also differ between desiccation-tolerant species, but a general response in resurrection pteridophytes includes high levels of ROS scavenging enzymes such as superoxide dismutases, catalases, peroxidases, and glutathione reductase. Indeed, it has been proposed that some desiccation tolerant *Selaginella* species are already primed for desiccation [20,83]. These desiccation-tolerant species exhibit significantly higher levels of compounds with antioxidant capacity compared to desiccation-sensitive relatives in normal conditions. Both tolerant and sensitive plants display an increase in antioxidant compounds upon dehydration, but the levels in sensitive species are generally lower than those observed in tolerant plants. The adaptive mechanism of priming that some resurrection plants exhibit has been proposed as an important component of VDT. For example, the desiccation-tolerant grass *Sporobolus stapfianus* is metabolically [88] and transcriptionally [89] primed for desiccation. While desiccation-sensitive species direct their metabolism to support faster growing rates, a desiccation-tolerant plant such as *S. stapfianus* invests part of its metabolism to resist dehydration and associated damage with the accumulation of osmolytes in non-stressful conditions.

Resurrection pteridophyte species can also deal with oxidative stress by the induction of secondary metabolism pathways (Table 2). A detailed analysis in *S. lepidophylla* identified the accumulation of several of these metabolites including flavonoids (e.g., apigenin), vanillate, and some phenylpropanoids (e.g., coniferyl alcohol) [65], all considered as potent antioxidants. A significant accumulation of sugar alcohols or polyols (mainly sorbitol and xylitol) is observed in *S. lepidophylla* compared to the desiccation sensitive *Selaginella moellendorffii* [83]. Polyols have a dual function participating in redox control and osmotic adjustment [90]. Due to the strong water-binding activity of polyols, they probably act to slow down water loss, providing a longer time for the induction and establishment of VDT mechanisms [65]. In response to dehydration, plants also accumulate compatible solutes (e.g., proline, mannitol, glycine betaine) to increase their cellular osmolarity and reduce water efflux from cells [91]. Proline accumulation in plants is a common response to several stresses, and in addition to its role as an osmolyte, it has been proposed to act as a chaperone preventing protein aggregation and as a ROS scavenger, among other functions [92]. The results of our survey showed that proline accumulation in response to desiccation is one of the most widespread mechanisms employed by desiccation-tolerant pteridophyte species (Table 1).

Among conserved responses in resurrection plants during dehydration, accumulation of LEA proteins and their transcripts to high levels is among the most frequent responses to water loss [47]. Surprisingly, LEA proteins are not restricted to plants and can be found in a wide range of organisms including animals that survive severe water stress [93]. The occurrence of LEA proteins in distinct life forms suggests an ancient and ubiquitous protective role in DT. Diverse functions have been proposed for LEA proteins including roles as molecular chaperones and the protection of cellular components [5]. However, characterization of the distinct LEA families showed multiple functions, interaction with proteins and nucleic acids and other activities, thus, apparently no single function is universal across LEA families [51]. As discussed above, several mechanisms involved in seed DT are also present in vegetative tissues of resurrection plants. For instance, *Arabidopsis* seeds acquired DT during the last stage of their development by accumulating high levels of LEA proteins and some sugars (i.e., sucrose and the raffinose family oligosaccharides) [34,37], which also accumulate in several desiccation-tolerant species. The interaction between both LEA and sugars has been proposed as the main constituents for the formation of intracellular glasses [94]. This glassy state, also known as vitrification, has been proposed as an important mechanism to relieve mechanical stress, reduce membrane fusion, and also in the inhibition of chemical reactions that could produce cellular damage [5]. A genome analysis of the lycophyte *S. lepidophylla* identified a total of 65 LEA genes, of which 48 showed significant expression changes during a rehydration–dehydration cycle [66]. Analysis of other resurrection lycophyte of the same genus, the species *Selaginella tamariscina*, also reported LEA genes as highly expressed during dehydration [95]. Most LEA genes in the desiccation-sensitive relative *S. moellendorffii* are also induced in response to dehydration [95]. The latter observation suggests a conserved protective role of LEA proteins on both DT and in response to dehydration in sensitive plants. Maybe the protection characteristics of LEA proteins during desiccation are due to its level of expression, the induction of particular members of some LEA families, or their combined activity with other compounds (e.g., sugars) produced during desiccation.

A decrease in the photosynthetic activity is a conserved response in all desiccation-tolerant plants [53]. Most of the physiological studies performed in resurrection pteridophytes have also described a decline in photosynthesis during dehydration (Table 1). Such a decrease is likely to be a consequence of a lower diffusion of CO_2_ caused by stomata closure and downregulation in the expression of photosynthesis-related genes, among other factors. An important difference between desiccation-tolerant and sensitive plants is that the former shuts down photosynthesis at early stages of dehydration to diminish oxidative damage [53]. The controlled cessation of photosynthesis also avoids the excessive production of ROS [52]. Although photosynthetic activity is arrested during dehydration, several photosynthesis related proteins including those involved in the maintenance of chloroplast stability and enzymes of the Calvin cycle are upregulated upon desiccation in some lycophytes [82]. Deactivation of photosynthetic activity at the proteome level has been reported in *S. tamariscina*, which shows a significant reduction in the abundance of key enzymes involved in CO_2_ fixation including the RuBisCO large subunit (rbcL) [78]. As discussed throughout this review, resurrection pteridophytes can show contrasting patterns for the same function. For example, the rbcL protein has been reported to remain constant and highly abundant during the desiccation process in some filmy ferns [68]. Apparently, regulation of photosynthesis in homoiochlorophyllous species occurs at the physiological, transcriptomic, and proteomic level, indicating a crucial role in the proper regulation of this process in the acquisition of VDT.

Additional to the mechanisms indicated in Table 1, desiccation-tolerant pteridophytes undergo morphological modifications during drying. Common mechanical responses to dehydration include leaf folding and stem curling, which has been proposed to diminish oxidative stress by reducing the area of photosynthetic tissue exposed to direct sunlight. A field experiment determined that stem curling in *S. lepidophylla* is a mechanism that limits photoinhibitory and thermal damage [96]. The ordered dehydration-induced morphological packing in *S. lepidophylla* is produced by a differential rate of water loss between the inner and outer stems [97]. A more detailed analysis showed the complexity of these mechanical processes, finding gradients of tissue density and even cell wall composition differences between the adaxial and abaxial sides of the stem [98]. Additionally, desiccation leads to a significant decrease in cell volume, causing mechanical stress that could be ameliorated by cell wall folding and vacuolation [5]. Structural characterization and transcriptomic analysis in *Selaginella* suggest a pronounced shift in cell wall structure and composition during desiccation [76,84].

## 5. Landscape of VDT Strategies in Pteridophytes

There is not a general conserved mechanism of VDT in plants, but some similarities as well as species-specific adaptations have been reported. Evidence of the diversity and plasticity of the mechanisms that confer DT in different plant lineages has previously been reported [5,7,11], but these studies are still incomplete, and the variety of mechanisms used to achieve VDT might turn out to be larger than initially thought. As discussed above, desiccation-tolerant pteridophytes display considerable ecological diversity, contrasting anatomical and physiological differences, distinct drying kinetics, and strategies to survive in the dried state. Furthermore, the ability to survive desiccation can also be seasonally regulated as reported for the fern *Mohria caffrorum,* which exhibits DT characteristics during the dry season but behaves as a sensitive plant in the rainy season [71].

There are obvious differences between shoots and roots including anatomical characteristics and the microclimate at which each tissue is exposed (direct drying by the surrounding air or through the rhizosphere, respectively). Therefore, it is expected that these two tissues differ in their responses to desiccation. Few studies have compared the response to desiccation between the shoot and root tissues. A study in *S. bryopteris* showed that the roots and shoots share some desiccation responses such as a similar change in energy metabolism, specifically an increase in the abundance of enzymes involved in ATP production [82]. However, some responses are specific or mainly activated in one of the two organs (organ-specific response). For instance, dehydration signal transduction in the roots triggers a stress response probably mediated by ABA, whereas in the shoot, a higher number of ROS scavenging proteins are activated for protection of the photosynthetic apparatus [82]. The increase in ABA levels is a well-conserved response to water stress conditions in plants including desiccation-tolerant species. Indeed, dehydration treatment in the lycophyte *S. tamariscina* caused an increment in ABA content of about three times compared to the hydrated conditions [79]. The same study demonstrated ABA-dependent expression of some genes involved in VDT including ELIP and LEA genes. Furthermore, a transcriptomic analysis carried out in the same species identified the key enzyme in ABA synthesis (NCED) as significantly induced under drought stress [85]. However, our knowledge about desiccation sensing and signaling pathways that are activated in resurrection plants is still limited. Current evidence indicates an important role of ABA regulation in plant responses to desiccation, but other hormones such as salicylic acid or ROS signaling are also likely to be involved in the acquisition of VDT [99].

Transcriptomic analyses provide the opportunity to study the dynamic changes in the gene expression of resurrection plants at different water status during the dehydration and rehydration process. The current available RNA-Seq data of desiccation-tolerant pteridophytes is limited to three filmy ferns and three lycophyte species (Table 2), but to our knowledge, there has been no transcriptomic analysis in resurrection ferns with well-developed cuticles. Compared to most desiccation-tolerant vascular plants, filmy ferns could experience very rapid drying rates and their recovery is also very fast (within ca. 30 min evaluated by net assimilation rates), suggesting constitutive VDT strategies [91]. Interestingly, filmy ferns can be subjected to de-acclimation (reduction in its tolerance) if stored in moist conditions for a relatively short period of time (one week) [30]. A similar phenomenon can be observed in the angiosperm *Boea hygrometrica*, which requires a period of acclimation to survive rapid desiccation [100], indicating that tolerance in filmy ferns also shares some features with the VDT observed in more complex plants. In accordance with the largely constitutive protection strategy proposed [68], only a few genes whose transcription is activated or repressed during the desiccation–rehydration process have been reported for filmy ferns [80,81]. The major changes in transcripts levels in the filmy fern *H. cruentum* take place upon rehydration, and this change is mainly on the activation of genes involved in ROS scavenging and photoprotective mechanisms (i.e., ELIPs) [81]. Furthermore, rehydration is the stage with most stress severity in filmy ferns during the whole desiccation process (indicated by a burst of lipoxygenase volatiles) [91]. Transcripts with high abundance regardless of the hydration state are related to translation, photosynthesis, and antioxidant activity, representing part of the putative constitutive mechanisms in filmy ferns [80]. These findings allow us to speculate that a high and steady expression of photosynthesis and antioxidant genes are directly involved in the adaptation to extreme water fluctuations in vascular plants.

The finding that VDT appeared repeatedly in phylogenetically distant plant lineages is indicative of convergent evolution of this ability [11]. In fact, this phenomenon can be observed even within members of the same lineage. In contrast to filmy ferns, desiccation tolerant *Selaginella* species display a much more dynamic gene expression response to desiccation. A comparative analysis showed that even closely related desiccation tolerant *Selaginella* species induce different sets of genes to activate similar mechanisms to survive desiccation, in addition to some species-specific adaptations. Although *Selaginella sellowii* and *S. lepidophylla* share most of their predicted protein families, most of the genes activated during dehydration and rehydration differ between these species [84]. Both species converged in the activation of some processes related to VDT including changes in amino acid metabolism, activation of antioxidant systems, and secondary metabolism. Because most genes are shared between these *Selaginella* species, the difference in the activation or repression of genes that play key roles in VDT derives from an event of the rewiring of regulatory networks that control the response to dehydration instead of the acquisition of novel genes. Transcriptomic analysis of *S. sellowii* and *S. lepidophylla* also uncovered some species-specific responses such as an unusual enrichment in the induction of photosynthetic genes during water loss in *S. sellowii*. During rehydration, *S. sellowii* has a faster recovery of photosynthesis compared to its relative *S. lepidophylla*, and this ecological advantage is likely to be related to higher expression of photosynthetic related genes. Additionally, a comparative analysis between two closely related angiosperms species of the genus *Lindernia* showed that VDT evolved through a combination of gene duplications and the rewiring of regulatory networks, regulating the expression of shared genes [101]. Together, these results represent an example of the broad landscape behind the gene regulatory networks controlling the activation of VDT.

Most plants have the genetic potential for DT, but most genes responsible for DTare only activated in reproductive structures. Genome analyses represent crucial resources to study the hidden potential to develop VDT. The genome of a quillwort with desiccation-tolerant corms (*I. taiwanensis*) was recently reported [58], but no data on its molecular responses to dehydration were reported. To date, among the pteridophytes whose genome has been sequenced, only for the lycophytes *S. lepidophylla* [66] and *S. tamariscina* [95] have data on their global transcriptional response to desiccation. Although there is no evidence of whole-genome duplication events in *Selaginella* [102], both genomes have signs of expansion of gene families related to DT traits. For instance, an expanded number of oleosin genes and structural proteins with protective roles including membrane repair are present in the *S. tamariscina* genome [95]. Gene duplication analysis indicated a major expansion of ELIP and LEA genes in the *S. lepidophylla* genome [66]. Interestingly, further analysis revealed that the ELIP gene family is expanded in all sequenced resurrection genomes compared to sensitive species [103]. Interestingly, over 74% of ELIP genes have been detected as clusters of continuous repeated genes in the genomes of desiccation-tolerant plants, indicating that most ELIP genes are derived from tandem gene duplications.

In conclusion, available evidence suggests that the dominant VDT strategy that a pteridophyte displays is directly correlated with its anatomical complexity. Specifically, VDT in pteridophytes with more efficient anatomies to retain water or to reduce drying rate is mainly based on dehydration-induced mechanisms whereas VDT in pteridophytes with a simpler leaf anatomical structure, which are predisposed to suffer faster drying rates, relies on a constitutive activation of the protection mechanism. We hypothesize that the faster the drying rate that a pteridophyte can tolerate, the larger the number of constitutive protection mechanisms it possesses. The schematic representation in Figure 3 is a summary of the main observations of a dominant constitutive or inducible desiccation strategy and some characteristic responses reported for desiccation-tolerant pteridophytes. As above-mentioned, a limited number of pteridophytes have been characterized despite the high incidence of VDT within this plant family. Further studies in resurrection pteridophytes will provide insights into the strategies that these plants evolved to survive desiccation.

## 6. Multiple Abiotic Stress Tolerance or a Consequence of Its DT Ability?

Different types of abiotic stresses cause similar disorders or damage inside cells. For instance, drought, salinity, and extreme temperatures cause oxidative stress due to excessive ROS accumulation [104]. Thus, it is possible that plants evolved protective responses that can be activated by multiple environmental signals. Most of the desiccation-tolerant plants inhabit tropical and subtropical regions of the world [48], however, a few species such as the angiosperms *Haberlea rhodopensis* and members of the genus *Ramonda* can withstand temperatures below the freezing point during winter [105,106]. Furthermore, the Maritime Antarctic is one of the environments with the most extreme conditions on our planet, and the moss *Sanionia uncinata* exhibits DT ability among the adaptations that allow it to grow under such extreme circumstances [107]. A recent study showed that the desiccation-tolerant ferns *Ceterach officinarum*, *Asplenium trichomanes,* and *Polypodium vulgare* also tolerate freezing, suggesting a relationship between desiccation and cold tolerance [108]. At environmental conditions, freezing leads to dehydration in *H. rhodopensis* until the leaves reach an air-dry state with the progression of winter [109]. Leaves of *H. rhodopensis* show rolling after exposure to subzero night temperatures, as occurs in response to desiccation. Leaf rolling seems to be caused by a water potential gradient when the temperature decreases, and that water can flow out of the leaves through narrow channels located at the epidermis [109]. Similarly, freezing induces frond-curl in the fern *P. vulgare* that is reversible upon thawing [108]. However, there are still no data to determine if this morphological change is a common response of resurrection plants from temperate and polar habitats. Some of the changes and putative mechanisms required for freezing tolerance have been described. To acquire freezing tolerance, a cold acclimation period seems essential before exposure to subzero temperatures (resembling natural field conditions). Then, at freezing conditions, the plants experience a rearrangement of the cell content, modification in the abundance of several of the major photosynthetic proteins, and the accumulation of protective compounds such as zeaxanthin, flavonoids, and anthocyanins [108,109,110,111]. Subsequently, enzymatic and nonenzymatic antioxidants have an important role to ensure plant recovery during rehydration [112].

Low-molecular weight compatible solutes are accumulated during the cold acclimation process of plants from temperate and polar climates. These compounds include sugars such as sucrose, glucose, raffinose, and trehalose as well as other osmolytes such as proline and glycine betaine [113]. The accumulation of osmolytes, in addition to other cellular changes including an increase in ROS scavenger enzymes (i.e., superoxide dismutase, catalases, peroxidases), prepare the plant to survive subsequent chilling or sub-zero temperatures [113]. All of the above-mentioned responses are also observed in resurrection plants during desiccation, suggesting common regulation between water stress and low temperatures. Some of the genes induced by cold have dehydration or ABA responsive elements in their promoter regions [114]. Indeed, several of the transcription factor families that regulate cold tolerance have also been described and proposed as putative regulators in VDT. For example, transcription factors with a crucial role in plant cold acclimation include the C-repeat binding factors (CBF), which are also referred to as dehydration-responsive element-binding proteins (DREBs) [115]. The induction of a significant number of stress-responsive genes is regulated by DREBs, which are expressed in a wide range of environmental conditions, particularly cold and drought stress, but also during salt and ABA treatment [116]. Evaluation of desiccation-tolerant plants in different stress conditions could answer the question of whether these overlapping responses share common components in their regulatory networks or if each stress condition evolved completely independent networks to activate similar sets of genes.

During freezing, water availability inside the cells is significantly reduced as it is during dehydration. When extracellular water freezes, it generates a difference in the water potential, leading to the movement of water out of the cells [117]. However, our knowledge is insufficient to define whether during freezing, the cells of desiccation-tolerant plants withstand partial dehydration or undergo a similar stress condition to a desiccated state. Further analysis could provide evidence about the relationship and mechanisms behind tolerance to multiple abiotic stresses. In their natural habitats, plants are usually exposed to multiple types of stresses rather than one, as is often studied under laboratory conditions. As previously mentioned, some desiccation-tolerant plants occupy sites where they are subjected to extreme conditions. For instance, resurrection plants actively growing in outcrops can be exposed to high temperatures and high light stress conditions. A widespread response in plants is the synthesis of flavonoids after UV exposure as plants can produce distinct types of flavonoids with different light absorption properties [118]. Flavonoids are among the protective compounds abundantly accumulated in some resurrection pteridophytes, suggesting a link between the accumulation of such “sunscreens” and the ability of resurrection plants to colonize sites with high levels of radiation. Evidently, resurrection plants could also experience other abiotic stresses than those described here. It is possible that resurrection pteridophytes display tolerance to different types of abiotic stress other than desiccation. Therefore, desiccation-tolerant plants represent important genetic sources for crop improvement to cope with climate change.

## 7. Concluding Remarks and Future Directions

Survival to almost complete loss of cellular water is not an easy task and only a reduced number of plant species have evolved unique and extraordinary strategies to survive in a desiccated state. Instead of a conserved and unique trait, DT can be achieved by a wide landscape of different adaptations and strategies that plants have evolved to survive desiccation. In accordance with the position of pteridophytes within land plant phylogeny representing a link between bryophytes and more complex plants, these plants exhibit an intermediate VDT strategy between the predominantly constitutive mechanisms in bryophyte plants and the dehydration-induced response in angiosperms. Whereas some desiccation-tolerant pteridophytes possess a very simple leaf anatomical structure with no control of water loss and a constitutive dominant VDT strategy (i.e., filmy ferns), other pteridophytes developed structural adaptations to prevent rapid water loss and their VDT is mainly based on dehydration-induced responses. However, there is evidence that these two strategies are not mutually exclusive and several desiccation-tolerant pteridophytes exhibit components of both VDT strategies. To decipher the evolution of this trait in plants, it is important to consider the intermediate strategy of pteridophyte species within the plasticity of the DT phenomenon.

Several questions regarding the origin and regulation of VDT remain unanswered. Future research can be focused on the transcriptional regulation during spore maturation and a detailed comparison with vegetative tissue response during dehydration to obtain insights into the origin of VDT in resurrection pteridophytes. Although the number of transcriptomic studies in resurrection plants has increased significantly in recent years, the characterization of the gene expression of desiccation-tolerant ferns with well-developed cuticles is still missing. This type of ferns display a dominant dehydration-induced VDT strategy, which means that they probably show a more dynamic or even completely different transcriptional response to dehydration than that described for filmy ferns. Furthermore, the study of desiccation-tolerant pteridophytes with a dominant constitutive strategy compared with species of induced-dehydration protection mechanisms could provide important clues about the regulation of VDT and the rewiring events that lead to activating diverse biochemical and cellular processes to survive in the dry state.

An important step toward achieving practical application is the validation of the key genes and proposed regulators for VDT. Due to the lack of desiccation-tolerant plants as model systems, few studies have evaluated the regulatory mechanisms that activate adaptive responses to extreme environments. Most VDT studies have been performed in the dicot *Craterostigma plantagineum* [119], a species for which gene transfer protocols have already been established [120]. Furthermore, a recent analysis in this species reported a low correlation between transcript and protein levels during a dehydration and rehydration cycle [121]. Such differences indicate a sophisticated posttranscriptional regulation that increases the complexity of the mechanisms involved in the activation of VDT. Several resurrection pteridophytes have small diploid genomes, but efficient transformation and gene editing protocols are still missing. Therefore, it is urgent to develop reproducible protocols for these two technologies to be able to use reverse genetic approaches to functionally study the genes involved in the VDT mechanisms. The development of high-quality genomic resources is crucial to decipher the regulatory mechanisms activated during desiccation that likely participate and confer tolerance to other abiotic stress conditions.

The goal of generating knowledge about VDT mechanisms besides understanding the evolution of plants is for its future use in crop improvement. Resurrection plants have evolved a myriad of mechanisms to survive desiccation, but these plants are usually not only exposed to water deficit, instead, they cope with multiple and even combined stress conditions. Although no economic relevant resurrection plants are known, knowledge on the common and alternative mechanisms to acquire VDT has potential use in new breeding strategies to enhance abiotic stress tolerance in crops.

## Figures and Tables

**Figure 1 plants-11-01222-f001:**
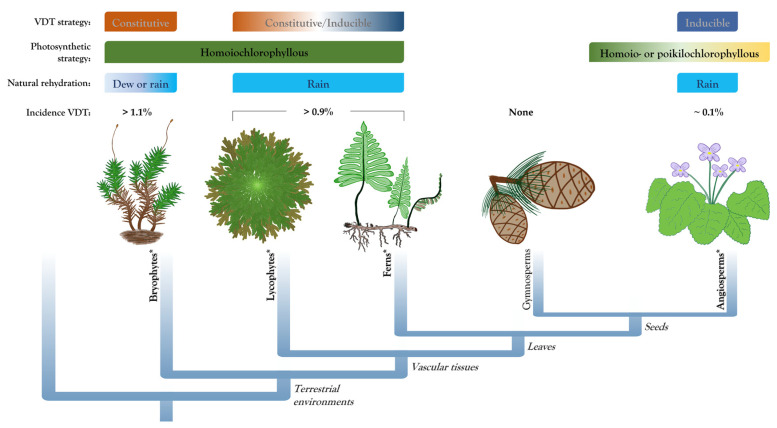
Desiccation tolerance strategies exhibited by land plants. The most common and widespread strategies for vegetative desiccation tolerance (VDT) in the major groups of land plants are indicated. Some additional desiccation related characteristics such as natural rehydration and estimated incidence of VDT in each group are also described. Clades with desiccation-tolerant members are indicated in bold and asterisks.

**Figure 2 plants-11-01222-f002:**
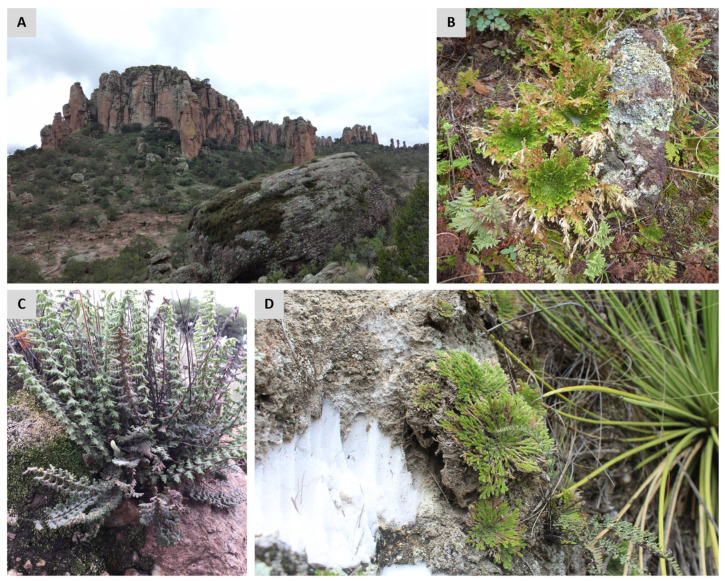
Common habitats for desiccation-tolerant pteridophytes. Photograph of an ecosystem with rock formations in the national park Sierra de Órganos, Mex. (**A**), and a close-up of a representative community of desiccation-tolerant organisms (including several ferns and mosses, *Selaginella* sp., lichens) growing on a rock outcrop (**B**). Resurrection plants can occupy rock crevices and shallow depressions where they experience periodic dryness, which represent inadequate sites for the establishment of desiccation-sensitive plants. Examples of desiccation-tolerant pteridophytes growing at these types of sites: the fern *Myriopteris aurea* (**C**) and the lycophyte *Selaginella pilifera* (**D**). All photographs were taken during the rainy season.

**Figure 3 plants-11-01222-f003:**
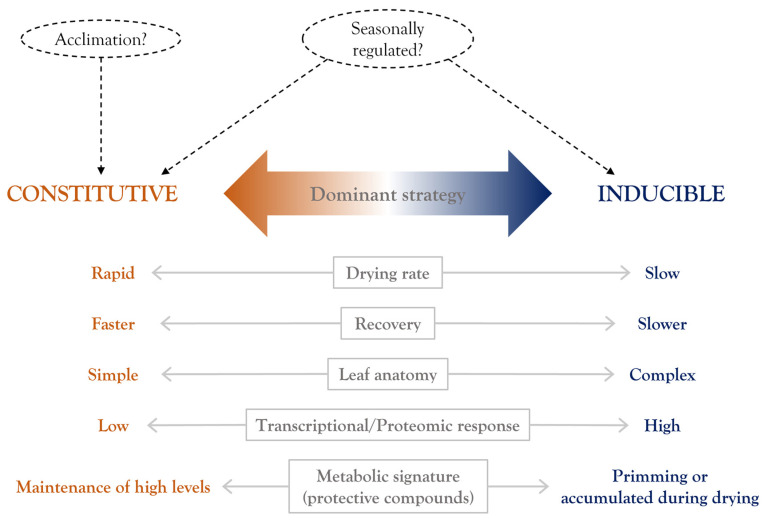
Overview of the intermediate vegetative desiccation tolerance strategy exhibited by pteridophytes. Desiccation-tolerant pteridophyte species can display a dominant strategy that is either constitutive (orange) or inducible (blue). A summary of the main characteristics and responses associated with each strategy are indicated. Some additional factors that can regulate desiccation tolerance capacity are indicated by discontinuous lines.

**Table 1 plants-11-01222-t001:** Biochemical and physiological responses in desiccation-tolerant pteridophytes.

Desiccation-Tolerant Species	Type	Response to Desiccation	Refs.
*Adiantum raddianum*	Fern	Sucrose ^DR^, proline ^DR^, SOD ^DR^, POD ^DR^, GR ^DR^, CAT ^DR^, partial Chl content decrease ^D^, superoxide radical increase ^D^, lipid peroxidation ^D^	[67]
*Crepidomanes inopinatum*	Filmy fern	Soluble sugars ^D^, SOD ^D^, **POD** ^C^, photosynthesis decline ^D^	[30]
*Hymenophyllum caudiculatum **	Filmy fern	**Sucrose**^C^, fatty acid composition ^DR^, photosynthesis decline ^D^	[68,69,70]
*Hymenophyllum cruentum **	Filmy fern	Significant Chl content decrease ^D^, photosynthesis decline ^D^	[31,69]
*Hymenophyllum dentatum **	Filmy fern	Sucrose ^D^, significant Chl content decrease ^D^, photosynthesis decline ^D^	[31,68,69]
*Hymenophyllum plicatum **	Filmy fern	Fatty acid composition ^DR^, photosynthesis decline ^D^	[69,70]
*Loxogramme abyssinica*	Fern	Soluble sugars ^D^, SOD ^D^, **POD** ^C^, photosynthesis decline ^D^	[30]
*Mohria caffrorum*	Fern	Fronds from dry compared to rainy season: **CAT**^C^, **GR**^C^, **SOD**^C^, sucrose ^D^, raffinose ^D^, vacuolation ^D^, photosynthesis decline ^D^	[71]
*Pleopeltis pleopeltifolia **	Fern	Proline ^R^, soluble sugars ^D^, significant Chl content decrease ^D^, carotenoid content decrease ^D^, photosynthesis decline ^D^	[57]
*Pleopeltis polypodioides* * (syn. *Polypodium polypodioides*)	Fern	CAT ^R^, fatty acids (linolenic, linoleic, palmitic and stearic acid) ^D^, CWF ^D^, LEA (dehydrin) ^D^, hydroperoxide content, and LPO increase ^D^	[72,73]
*Selaginella brachystachya*	Lycophyte	Anthocyanin ^D^, POD ^DR^, CAT ^D^, SOD ^DR^, GR ^DR^, proline ^D^, sucrose ^DR^, partial Chl content decrease ^D^, carotenoid content decrease ^D^, LPO increase ^D^, superoxide radical increase ^D^, stomata closure ^D^, photosynthesis decline ^D^	[74]
*Selaginella bryopteris*	Lycophyte	Proline ^DR^, SOD ^DR^, APX ^DR^, CAT ^D^, photosynthesis decline ^D^, reduced stomatal conductance ^D^	[75]
*Selaginella involvens*	Lycophyte	CWF ^D^, modifications in cell wall composition ^D^	[76]
*Selaginella lepidophylla*	Lycophyte	CWF ^D^, vacuolation ^D^, sucrose ^D^, increased flavonoid and phenol content ^D^, photosynthesis decline ^D^, partial Chl content decrease ^D^	[20,62,77]
*Selaginella sellowii*	Lycophyte	Increased flavonoid and phenol content ^D^, photosynthesis decline ^D^, partial Chl content decrease ^D^	[20]
*Selaginella tamariscina*	Lycophyte	ABA ^DR^, proline ^D^, soluble sugars ^DR^, SOD ^DR^, POD^D^, GR ^DR^, CAT ^DR^, ELIPs ^D^, LEA ^D^, **high levels of trehalose ^C^, low saturation ratio of phospholipids** ^C^, photosynthesis decline ^D^, reduced stomatal conductance ^D^, partial Chl content decrease ^D^	[78,79]

The protection mechanisms are indicated by the enzyme, gene/protein, or compound that showed higher activity, expression, or accumulation, respectively, during dehydration (superscript ^D^), rehydration (superscript ^R^), or both (superscript ^DR^) compared to hydrated conditions. Constitutive mechanisms (in **bold** and superscript ^C^) are proposed when a high level was reported and no statistical difference between hydrated conditions and desiccation treatment was determined. Some physiological responses with a change in relation to tissue in hydrated conditions are also listed. Photosynthesis decline was determined by net assimilation rate, CO_2_ exchange rate, or indirectly using photosynthetic parameters (most of the studies used *Fv/Fm* measurements). Epiphytic species are indicated with an asterisk (*). ABA, abscisic acid; APX, ascorbate peroxidase; CAT, catalase; Chl, chlorophyll; CWF, cell wall folding; ELIPs, early light-inducible proteins; GR, glutathione reductase; LEA, late embryogenesis abundant; LPO, lipid peroxidation; POD, peroxidase; SOD, superoxide dismutase.

**Table 2 plants-11-01222-t002:** A summary of the desiccation responses identified by transcriptomic, proteomic, and metabolic approaches in desiccation-tolerant pteridophytes.

Desiccation-Tolerant Species	Type	Desiccation Tolerance Mechanisms	Refs.
*Hymenophyllum caudiculatum **	Filmy fern	Proteomic analysis: Few differences in protein patterns between hydration states suggest constitutive expression. Transcriptomic analysis: Few transcripts differential expressed suggesting constitutive strategy. Specific responses include osmotic and phenylpropanoid pathways.	[68,80]
*Hymenoglossum cruentum **	Filmy fern	Transcriptomic analysis: Few transcripts differential expressed suggesting constitutive strategy. The major expression change occurs during rehydration.	[81]
*Hymenophyllum dentatum **	Filmy fern	Proteomic analysis: Few differences in protein patterns between hydration states suggest constitutive expression. Transcriptomic analysis: Few transcripts differential expressed suggesting constitutive strategy. Specific responses include oxidative damage and high light stress.	[68,80]
*Selaginella bryopteris*	Lycophyte	Proteomic analysis: Stress and defense, carbohydrate and energy metabolism. *Shoot specific:* Photosynthesis protection, protein metabolism. *Root specific:* Nucleotide metabolism, signaling	[82]
*Selaginella lepidophylla*	Lycophyte	Metabolic analysis: Constitutive strategy or predisposition to desiccation in: carbohydrate metabolism (sucrose, mono- and polysaccharides), sugar alcohols (e.g., sorbitol, xylitol). Additional responses include changes in: amino acids (aromatic aa), glutathione metabolism, secondary metabolites (flavonoids). Transcriptomic analysis: Upregulated transcripts included amino acid metabolism, carbohydrate metabolism (mainly trehalose), cell wall modification, antioxidant system (peroxiredoxin, dismutases and catalases), secondary metabolism (flavonoids, phenylpropanoids, isoprenoids), transport (MIPs).	[83,84]
*Selaginella sellowii*	Lycophyte	Transcriptomic analysis: Upregulated transcripts included amino acid metabolism, carbohydrate metabolism (mainly raffinose family), photosynthesis (Calvin cycle and lightreactions), antioxidant system (peroxiredoxin), secondary metabolism (flavonoids, phenylpropanoids), transport (MIPs).	[84]
*Selaginella tamariscina*	Lycophyte	Proteomic analysis: Most proteins are downregulated upon dehydration. Upregulated proteins belong to carbohydrate and energy metabolism, protein metabolism/modificantion (HSP/chaperonines), ELIPs, LEA proteins. Transcriptomic analysis: More upregulated than downregulated genes were identified. Upregulated genes included antioxidant system (POD), cell wall modification, osmotic adjustment, cuticle defense (biosynthesis), LEA proteins.	[78,79,85]

The main findings of each study are described indicating the proposed protection mechanisms. For significantly enriched categories or pathways, some outstanding compounds or processes are indicated in brackets. Epiphytic species are indicated with an asterisk (*). ELIPs, early light-inducible proteins; HSP, heat shock proteins; LEA, late embryogenesis abundant; MIPs, major intrinsic proteins; POD, peroxidase.

## Data Availability

Not applicable.

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
