# Peer review of "Exploring the High Variability of Vegetative Desiccation Tolerance in Pteridophytes"

_plants, 2022, doi:10.3390/plants11091222_

Round 1

Reviewer 1 Report

I write this review as a botanist well acquainted with desiccation tolerant pteridophytes in nature.

I found this a well constructed review which will be of general interest . Even though no work seems to have been done on these, the survival of geophytic Isoetes through dry seasons should be mentioned.

 The English is somewhat  awkward  with numerous simple grammatical errors. It  would benefit from minor editing and shortening. Tone down references to pteridophytes being intermediate between bryophytes and seed plants.

The abstract is rather uninformative It could be rewritten to  highlight better the key features of pteridophyte  desiccation biology.

Author Response

Reviewer 1

We thank the reviewer for taking the time to critically review and comment our manuscript. We found the suggestion of including desiccation a small section on desiccation tolerance  in Isoetes important. Therefore, in the revised version of this review we incorporated the following paragraphs on what is known on desiccation tolerance in Isoetes:

Comment 1: Even though no work seems to have been done on these, the survival of geophytic Isoetes through dry seasons should be mentioned.

Author response: We thank the reviewer for this important suggestion. As the reviewer aptly indicated in this comment, the characterization of desiccation tolerance ability in Isoetes has been poorly studied. Despite a very limited number of studies have describe Isoetes desiccation strategy, we found that this group of plants evolved a particular way to survive desiccation and it is important to discuss in the present review. In this revised version of our manuscript, we included a paragraph and several statements contrasting the desiccation tolerance strategy reported in Isoetes compared to other pteridophyte species.

Line 103 to 110: “The ability to survive desiccation is also present in Isoetes, commonly known as quillworts, but knowledge on their responses to desiccation and mechanisms of DT have been little studied. Although Isoetes species most often occupy aquatic or semiaquatic habitats, some of them experience occasional drought. In contrast to other resurrection pteridophytes, the VDT in Isoetes is apparently restricted to corms [21], underground stem structures that acts as a food-storage structure. The dried corms of the species Isoetes taiwanensis can remain viable for several months, and few days after rehydration they produce new leaves and roots [22]. Due to the temporal variability in water availability in the habitats of some Isoetes, it is possible that several of these species could display VDT ability.”

Line 296 to 303: “Interestingly, desiccation-tolerant corms of Isoetes can neither be classified as homoiochlorophyllous nor poikilochlorophyllous species, because its leaves (and also roots) decay and lose function after desiccation, and their corms develop completely new photosynthetic tissue upon rehydration [22]. The particular way that these species evolved to deal with desiccation increase the diversity of VDT strategies observed in pteridophytes. Furthermore, I. taiwanensis also exhibits crassulacean acid metabolism (CAM) [58] providing the opportunity to study the crosstalk between a strategy to improve water-use efficiency, and the ability to tolerate extreme dehydration.”

Comment 2: The English is somewhat awkward with numerous simple grammatical errors. It would benefit from minor editing and shortening.

Author response: We thank the reviewer for this observation. We identified and corrected grammatical errors in this revised version of our manuscript.

Comment 3: Tone down references to pteridophytes being intermediate between bryophytes and seed plants.

Author response: We now make clear that pteridophytes have a range of different desiccation tolerance mechanisms  and cannot be simply considered as intermediates between bryophytes and seed plants.

Comment 4: The abstract is rather uninformative It could be rewritten to highlight better the key features of pteridophyte desiccation biology.

Author response: We thank the reviewer for this comment. We modified the abstract in the revised version of our manuscript including statements of each one of the most important sections of the present review.

Modified abstract: “In the context of plant evolution, pteridophytes, which comprises lycophytes and ferns, occupy an intermediate position between bryophytes and seed plants sharing characteristics with both groups. Pteridophytes is a highly diverse group of plant species that occupy a wide range of habitats including ecosystems with extreme climatic conditions. There is a significant number of pteridophytes that can tolerate desiccation by temporarily arresting their metabolism in the dry state and reactivating it upon rehydration. Desiccation-tolerant pteridophytes exhibit a strategy that is intermediate between the constitutive and inducible desiccation tolerance (DT) mechanisms observed in bryophytes and angiosperms, respectively. In this review, we first describe the incidence and anatomical diversity of desiccation-tolerant pteridophytes, and discuss recent advances on the origin of DT in vascular plants. Then, we summarize the highly diverse adaptations and mechanisms exhibited by this group, and describe how some of these plants could exhibit tolerance to multiple types of abiotic stress. Research on the evolution and regulation of DT in different lineages is crucial to understand how plants adapted to extreme environments. Thus, in the current scenario of climate change, the knowledge of the whole landscape of DT strategies is of vital importance as a potential basis to improve plant abiotic stress tolerance.”

Reviewer 2 Report

Manuscript ID: plants-1702330

Title: Exploring the high variability of vegetative desiccation tolerance in pteridophytes

Authors: Gerardo Alejo-Jacuinde and Luis Herrera-Estrella

Recommendation: Minor revision

The phenomenon of extreme desiccation is rarely spread into the Plant kingdom. The authors review the mechanism of desiccation tolerance of Pteridophytes, which share common mechanism of vegetative desiccation tolerance, which are characteristic of both bryophytes and angiosperms. The review is important because lycophytes and ferns species are the “link” between constitutive and inducible tolerance to extreme dehydration. There is still much unknown about the process of desiccation tolerance and the mechanisms that plants use to cope with extreme water loss.

The review is very well written. It is not easy to compare the mechanisms of desiccation tolerance used by all types of resurrection plants, taking in mind their various specie-specific responses to desiccation. Discussion about cross-tolerance is well appreciated. You can find my comments below.

Comments:

What I miss in the manuscript is the review of the photosynthetic process of resurrection plant (downregulation of CO2 fixation and chlorophyll fluorescence, changes in the amount of photosynthetic proteins). Maybe the authors could write a little paragraph about it.

Line 327: Table 1 – “Physiological and biochemical responses in desiccation-tolerant pteridophytes.” – I can see only the biochemical responses in Table 1, the physiological ones are missing. Please, correct the table and include the physiological responses.

Line 332: “arly light-inducible proteins” – change to “early”

Line 557: 6. Multiple abiotic stress tolerance or a consequence of its DT ability? – Extreme habitat conditions are characteristic of all resurrection species. Is interesting to investigate the cross-tolerance of the plants and to compare the shared mechanisms used to cope with different stress factors. In addition to being exposed to low temperatures, resurrection plants are also exposed to low or high light, or high temperatures. Maybe the authors could add something about it.

Line 593: write “,” instead of “.”

Author Response

Reviewer 2

We thank the reviewer for taking the time to critically review and comment our manuscript. We agree with all the comments and suggestions. The recommendation to add information on photosynthesis regulation, additional abiotic stresses, and physiological responses in resurrection pteridophytes undoubtedly improve our manuscript.  

Comment 1: What I miss in the manuscript is the review of the photosynthetic process of resurrection plant (downregulation of CO2 fixation and chlorophyll fluorescence, changes in the amount of photosynthetic proteins). Maybe the authors could write a little paragraph about it.

Author response: We agree with this and have incorporated your suggestion. Due to the different and even contrasting photosynthetic patterns observed in some desiccation-tolerant pteridophytes, it is difficult to discuss in detail this topic in this review. Nevertheless, we now describe the generally observed decline in photosynthetic activity in resurrection plants and the physiological processes likely associated with such decrease. Lastly, we mentioned an example contrasting patterns in photosynthetic activity observed in pteridophytes.

Line 393 to 409: “A decrease in the photosynthetic activity is a conserved response in all desiccation-tolerant plants [53]. Most of the physiological studies performed in resurrection pteridophytes have also described a decline in photosynthesis during dehydration (Table 1). Such decrease is likely a consequence of a lower diffusion of CO2 caused by stomata closure and downregulation in the expression of photosynthesys-related genes, among other factors. An important difference between desiccation-tolerant and sensitive plants, is that the formers shut down photosynthesis at early stages of dehydration to diminish oxidative damage [53]. The controlled cessation of photosynthesis also avoids the excessive production of ROS [52]. Although photosynthetic activity is arrested during dehydration, several photosynthesis related proteins including those involved in maintenance of chloroplast stability and enzymes of the Calvin cycle are upregulated upon desiccation in some lycophytes [79]. Deactivation of photosynthetic activity at proteome level has been reported in S. tamariscina, which shows a significant reduction in the abundance of key enzymes involved in CO2 fixation including RuBisCO large subunit (rbcL) [80]. As discussed throughout this review, resurrection pteridophytes show contrasting patterns for the same function. For example, rbcL protein has been reported to remain constant and highly abundant during the desiccation process in some filmy ferns [81]. Apparently, regulation of photosynthesis in homoiochlorophyllous species occurs at the physiological, transcriptomic and proteomic level indicating a crucial role of the proper regulation of this process in the acquisition of VDT.”

Comment 2: Table 1 – “Physiological and biochemical responses in desiccation-tolerant pteridophytes.” – I can see only the biochemical responses in Table 1, the physiological ones are missing. Please, correct the table and include the physiological.

Author response: We thank the reviewer for this important observation. We added physiological responses reported for each of the species listed in Table 1. Such physiological responses are described throughout the manuscript.

Comment 3: “arly light-inducible proteins” – change to “early”.

Author response: We thank the reviewer for this observation. We corrected such error in this revised version of our manuscript.

Comment 4: Multiple abiotic stress tolerance or a consequence of its DT ability? – Extreme habitat conditions are characteristic of all resurrection species. Is interesting to investigate the crosstolerance of the plants and to compare the shared mechanisms used to cope with different stress factors. In addition to being exposed to low temperatures, resurrection plants are also exposed to low or high light, or high temperatures. Maybe the authors could add something about it.

Author response: We fully agree with the reviewer, therefore, we incorporated a paragraph on cross-tolerance to different types of abiotic stress. Unfortunately there is not much information about tolerance to abiotic stress other than desiccation in resurrection plants..

Line 580 to 589: “As mentioned before, some desiccation-tolerant plants occupy sites where they are subjected to extreme conditions. For instance, resurrection plants actively growing in outcrops can be exposed to high temperatures and high light stress conditions. A widespread response in plants is the synthesis of flavonoids after UV exposure, plants can produce distinct types of flavonoids with different light absorption properties [112]. Flavonoids are among the protective compounds abundantly accumulated in some resurrection pteridophytes, suggesting a link between the accumulation of such “sunscreens” and the ability of resurrection plants to colonize sites with high levels of radiation. Evidently, resurrection plants could also experience other abiotic stresses than those described here. It is possible that resurrection pteridophytes display tolerance to different types of abiotic stress other than desiccation.”
